# Metabolomic Profiling of Female Mink Serum during Early to Mid-Pregnancy to Reveal Metabolite Changes

**DOI:** 10.3390/genes14091759

**Published:** 2023-09-04

**Authors:** Yuxin Luo, Yiqiu Huang, Liang Deng, Zheng Li, Chunjin Li

**Affiliations:** College of Animal Science, Jilin University, Changchun 130062, China; luoyx23@mails.jlu.edu.cn (Y.L.); 17843103601@163.com (Y.H.); chndengliang@163.com (L.D.); zhenglee19@163.com (Z.L.)

**Keywords:** mink, metabolomic profiling, amino acid, embryonic diapause

## Abstract

Mink embryos enter a period of diapause after the embryo develops into the blastocyst, and its reactivation is mainly caused by an increase in polyamine. The specific process of embryo diapause regulation and reactivation remains largely unexamined. This study aimed to identify changes in metabolites in the early pregnancy of mink by comparing and analyzing in serum metabolites up to twenty-nine days after mating. Blood samples were taken on the first day of mating, once a week until the fifth week. Metabolomic profiles of the serum samples taken during this period were analyzed by ultra-performance liquid chromatography/mass spectrometry. Multivariate statistical analyses identified differential metabolite expression at different time points in both positive and negative ion modes. The levels of dopamine, tyramine, L-phenylalanine, L-tyrosine, tyrosine, L-kynurenine, L-lysine, L-arginine, D-ornithine, and leucine changed significantly. These metabolites may be associated with the process of embryo diapause and subsequent reactivation.

## 1. Introduction

The American mink (*Neovison vison*) exhibits seasonal estrus and mating-stimulated ovulation. Mink embryonic diapause occurs during the blastocyst stage, a period of relatively static developmental state in embryonic development and metabolism [1]. The birth of offspring is delayed by the delayed implantation of blastocysts, allowing them to be born during favorable environmental conditions in terms of temperature and food availability. This reproductive strategy enhances the survival rate of the offspring.

From the vernal equinox, the photoperiod increases gradually, stimulating prolactin secretion from the pituitary gland [2,3]. Prolactin is essential for the reactivation of the embryo. When the secretion of prolactin increases, the development of the corpus luteum is initiated, which, in turn, promotes an increase in progesterone (P4) secretion [4]. When the concentration of P4 in the body reaches the requisite level for the attachment of the blastocyst to the uterine wall, the blastocysts initiate implantation and proceed to the fetal developmental phase [5]. During diapause, the embryo experiences a significant reduction in its metabolism, leading to minimal mitosis and only basic protein synthesis. The embryo is thus in a free state within the uterus [6].

The uterus is a crucial environment for embryonic development, including diapause. Therefore, both the uterus and intrauterine substances are essential for inducing and reactivating dormant embryos. A recent study has confirmed the impact of polyamines on diapause in mink embryos. Treatment of mink uterine epithelial cells with varying doses of prolactin demonstrated that this hormone prompts the expression of polyamine regulatory genes, such as ODC1 in the uterus, through the pSTAT1 and mTOR pathways, which in turn regulates the levels of polyamines in the uterus [7]. Putrescine is the initial polyamine molecule in the de novo polyamine synthesis tract found in all higher mammalian cells. It is significant in encouraging cell division, differentiation, the creation of macromolecules, such as DNA, RNA, and proteins, and the mitigation of inflammatory processes [8]. To determine whether polyamines can alleviate embryo diapause, embryos in diapause were incubated in vitro with putrescine. This resulted in an increase in blastocyst volume and the total number of cells, as indicated by further development of the embryos. However, the non-treated control embryos remained in a diapause state [7,9,10]. Together, these results provide strong evidence showing that embryos are maintained in a diapause state because of a restricted supply of uterine-derived polyamines.

A distinction in gene expression has been noticed between mink embryos in diapause and reactivation modes. More than 200 embryonic genes were up-regulated during activation. Concerning reactivation, there was an increase in the expression of embryonic genes associated with polyamine synthesis. Approximately 14% of the sequences that corresponded to the characterized genes were associated with the cell cycle, and 14% were associated with metabolism [7]. Therefore, it can be deduced that metabolism may have significance in the process of embryonic diapause and reactivation. Moreover, currently, no research has explored the alterations in the metabolome of minks pre and post embryo diapause.

Metabolomics is a crucial aspect of systems biology. It is the scientific discipline that explores the connections between endogenous metabolites and internal or external factors in whole organisms, systems, organs, and cells [11]. Metabolomic analysis techniques consist of gas or liquid chromatography coupled to mass spectrometry (GC-MS/LC-MS), infrared spectroscopy (IR), and nuclear magnetic resonance (NMR). Among these technologies, LC/MS has the advantages of high resolution, high sensitivity, and reproducibility. Consequently, it has been widely used for the identification and quantification of metabolites [12]. It is used to identify and quantify hundreds of metabolites with high mass accuracy for comprehensive metabolic profiling [13,14].

A pre-implantation embryo exists independent of blood supply and relies on energy sources from its in vivo environment (e.g., oviduct and uterine fluid) to sustain its development. Survival within this aqueous environment is possible due to the presence of amino acids, proteins, lactate, pyruvate, oxygen, glucose, antioxidants, ions, growth factors, hormones, and phospholipids [15]. Gas chromatography-mass spectrometry was employed to examine the metabolite profiles of the uterus, uterine fluid, and maternal plasma at the stages of pre-implantation and implantation. The research concludes that between pre-implantation day 1 and day 4, noteworthy metabolic variations were identified in the uterine fluid that may be of significance for blastocyst development and safeguarding against the tough uterine environment. In addition, the metabolites present in maternal plasma exhibited comparable alterations. For example, cysteine enhances the development and viability of blastocysts. These findings highlight the biochemical modifications observed in the maternal in vivo environment, which is necessary for meeting the dynamic demands of both the pre-implantation embryo and receptive endometrium [16]. Here, we investigated alterations in the serum extracts’ chemical composition throughout the distinct stages of pregnancy. As far as we know, no previous research has assessed the metabolic changes from fertilization through embryo diapause, from embryo diapause to embryo reactivation, during development, and during implantation. Our research aims to aid scientific advancement in this crucial field.

## 2. Materials and Methods

### 2.1. Animals and Serum Collection

All experiments on minks were performed under the direction and approval of the Animal Protection and Utilization Committee (SY202003300). Twenty multiparity minks (average age 3.5 years old, average weight 850 g) were farmed in the Special Economic Animal Experiment Base. Minks mate on 15th March and give birth on 26th April to produce offspring. Minks were exposed to natural light throughout the gestation period. All minks were raised under conditions of controlled temperature (22~24 °C) and humidity (60~70%), where they were provided with sufficient food (mink-specific feeds, including animal feeds, plant feeds, and supplementary feeds) and water. Each female mink of normal fertility mated with a fertile male mink by artificial insemination. Venous blood from the 20 female minks was taken one day (D1), eight days (D8), 15 days (D15), 22 days (D22), and 29 days (D29) after mating. The blood was marked and transported at 2–8 °C to the laboratory (tube does not contain anticoagulant) and centrifuged at 15,000× *g* for 15 min, and the serum was stored at –80 °C until analysis. The 20 minks were fed and observed continually after blood collection until the end of pregnancy. Of the 20 minks used, 15 subsequently gave birth. Six serum samples from mink of the same gestation period (41 days) were ultimately selected. Of the six mink sera from each sampling day, two sera with similar genetic backgrounds were mixed, and the six sera were used as three biological replicates for subsequent metabolomic profiling.

### 2.2. Sample Preparation

A serum sample (100 μL) was placed in an EP tube, to which 300 μL of methanol and 20 μL internal standard substances were added, followed by vortexing for 30 s. The samples were treated with ultrasound for 10 min (while incubated in ice water) and then incubated for 1 h at −20 °C to precipitate proteins. The sample was then centrifuged at 15,000× *g* for 15 min at 4 °C. The supernatant (200 μL) was transferred into a fresh 2 mL LC/MS glass vial, and 20 μL was taken from each sample and pooled together as quality control (QC) samples. An additional 200 μL of supernatant was taken for UHPLC-QTOF-MS analysis.

### 2.3. LC/MS Analysis

Metabolomic analysis was performed using an Agilent 1290 ultra-performance LC system coupled to quadrupole time-of-flight (Triple TOF 5600 AB Sciex) mass spectrometer, which is capable of performing primary and secondary mass spectrometry data acquisition based on the IDA function under the control of Analyst TF 1.7 software (AB Sciex). A UPLC BEH Amide column (1.7 μm; 2.1 × 100 mm) from Waters (Taunton, MMA, USA) was used for chromatographic separation; the column temperature was maintained at 40 °C. The LC-MS system was run in a binary gradient solvent mode. Solvent A contained 25 mM ammonium acetate and 25 mM ammonium hydroxide in water (pH 9.75), and solvent B contained 0.1% formic acid in acetonitrile. The flow rate was 500 μL/min. The linear gradient was as follows: 0 min, 95% B; 7 min, 65% B; 9 min, 40% B; 9.1 min, 95% B; 12 min, 95% B. Sample analysis was performed using the positive or negative electrospray ionization (ESI) mode; the injection volume was 3 μL. In each cycle, twelve precursor ions with an intensity greater than 100 were chosen for fragmentation at collision energy (CE) of 30 V (15 MS/MS events with a production accumulation time of 50 ms each). The ESI ion source parameters were set as follows: ion source gas 1 at 60 psi, ion source gas 2 at 60 psi, curtain gas at 35 psi, source temperature 650 °C, and ion spray voltage floating (ISVF) 5000 V or −4000 V in positive or negative modes, respectively. QC samples, which were prepared by mixing equal volumes (20 μL) from each serum sample as they were aliquoted for analysis, were used to assess the reproducibility and reliability of the LC-MS system.

### 2.4. Data Processing and Pattern Recognition

Unprocessed mass spectrometry data were converted to mzXML using ProteoWizard software and then processed with the R package (XCMS v3.2.). The pre-processing results generated a data matrix that consisted of the retention time (RT), mass-to-charge ratio (*m*/*z*) values, and peak intensity. The R package CAMERA was used for peak annotation after XCMS data processing.

The resulting scaled datasets were applied to the principal component analysis (PCA), which was used to validate the quality of the analytical system performance and to observe possible outliers. The orthogonal projections to latent structures-discriminant analysis (OPLS-DA) was used to analyze the results using SIMCA-P v11.0 software (Umetrics AB, Umea, Sweden) to filter out orthogonal variables in metabolites that are not related to categorical variables, analyze non-orthogonal variables and orthogonal variables separately to obtain an overview of the complete dataset, and discriminate between variables that are responsible for variation between the groups. OPLS-DA score plots were used to evaluate the quality of the model by the relevant R2 and Q2 parameters. The differential metabolites were selected by the combination of the *p*-value of the ANOVA and variable importance in the projection (VIP) values of the OPLS-DA model. The results are considered statistically significant as the *p*-value was less than 0.05, and the VIP value was larger than 1. The log2-fold change was used to show how these selected differential metabolites varied between groups. Metabolites that varied were also shown by volcano plot. The online database KEGG (http://www.genome.jp/kegg/ (accessed on 26 July 2021)) was used to annotate the potential differential metabolites by searching for the exact molecular mass data from redundant *m*/*z* peaks against a specific metabolite. Metabolomic analyses were performed using SPSS v22.0 software.

## 3. Results

### 3.1. Original Chromatogram Based on LC/MS

All sample analyses were performed in positive and negative ESI mode by using ultra-performance LC quadrupole time-of-flight tandem MS (UPLCQ-TOF-MS). A total of 849 metabolites were detected in positive ion mode and 853 metabolites in negative ion mode (Appendix A). A broad overlapping spectrum of QC samples showed that the instrument has repeatability and stability (Figure 1).

### 3.2. Multivariate Statistical Analysis

PCA and OPLS-DA were performed on the data from the 15 samples (five stages × three biological replicates) to observe differences in metabolic profiles among different time points. The analysis showed that there was a clear separation between D1 and another group in the PD1 × PD8 score plots (Figure 2).

To further search for ion peaks that discriminated between the two groups, a supervised OPLS-DA model was established because it focused on actual class discrimination more than the unsupervised PCA model. As shown in Figure 3, all R2X were greater than 0.5, R2Y was 1 or 0.999, and Q2 was greater than 0.9, except D1 vs. D8, in positive ion mode, indicating that the OPLS-DA model had a high predictive ability.

The OPLS-DA models discriminated adequately between different groups, both in positive and negative ion modes, which indicated that their metabolic characteristics were distinct. To validate the reliability of the OPLS-DA model, an alignment verification was performed (Figure 4). R2′ and Q2′ were lower than R2 and Q2 of the original model, which implies that the corresponding points did not exceed the corresponding lines, indicating that the model was meaningful. Thus, the differential metabolites can be screened according to the VIP.

### 3.3. Identification of Differential Metabolites and Metabolic Pathways

In the first instance, metabolites with a VIP value greater than 1 in the OPLS-DA model were selected. The ANOVA was used to select metabolites with significant changes (*p* < 0.05) (Appendix A). A volcano plot revealed differences in expression levels of the metabolites in the two groups and the metabolites with statistically significant differences (Figure 5).

A total of 33 and 49 differential metabolites of the comparison group 1 (D1 vs. D8) could be annotated in positive mode and negative mode using the KEGG pathway database (Appendix A). Most of the differential amino acids annotated were down-regulated, including L-proline, L-threonine, taurine, L-leucine, D-ornithine, L-lysine, L-valine, L-kynurenine, and L-glutamine (Table 1). On the contrary, L-methionine, L-phenylalanine, and L-tyrosine were up-regulated in D8 as compared to D1. Similarly, dopamine and its precursor, tyramine, were significantly increased in D8 as compared to D1 (Figure 6).

According to the KEGG pathway database (Appendix A, Table 2), 38 metabolites were annotated in positive mode and 61 in negative mode as differential metabolites of group 2 (D1 vs. D15), more than half of which were the same for group 1 (D1 vs. D8). Levels of tyrosine and dopamine in D8 and D15 were significantly higher than in D1. Almost all types of amino acids that are up-regulated or down-regulated are the same when compared to D8 and D15 with D1.

Regarding group 3 (D1 vs. D22), 45 differential metabolites in positive mode and 50 in negative mode were annotated (Appendix A, Table 3). As for group 4 (D1 vs. D29), 31 and 38 differential metabolites, respectively, in the positive and negative ion modes were members of metabolic pathways (Appendix A, Table 4). All amino acids were down-regulated on D22 as compared to D1, except L-methionine, L-tyrosine, L-phenylalanine, and dopamine. All amino acids were down-regulated on D29 as compared to D1, except L-phenylalanine. In addition, a further comparative analysis was conducted for D8 vs. D15 and D15 vs. D22 with PCA and OPLS-DA. The data showed that there was a clear separation between D8 and D15 and between D15 and D22 in the PD1 × PD8 score plots (Figure 7).

For a more in-depth search for ion peaks that discriminated between the two groups, the supervised OPLS-DA model was established. As shown in Figure 8, the OPLS-DA model was highly predictive. To validate the reliability of the OPLS-DA model, an alignment verification was performed (Figure 9). R2′ and Q2′ were smaller than R2 and Q2 of the original model, indicating that the model was meaningful.

Differential metabolites could be screened according to the VIP. Metabolites with a VIP value greater than 1 in the OPLS-DA model were selected, and the AVONA was used to select metabolites with significant changes (*p* < 0.05). Because many amino acids promote mouse embryo development and blastocyst activation [17,18,19], we focus on showing the changes in amino acids and dopamine in different groups. Dopamine was significantly higher on D15 than on D8, and the tyramine and L-tyrosine, precursors of dopamine, were also higher on D15 than on D8, but L-glutamine and D-ornithine decreased. L-lysine, L-arginine, L-kynurenine, DL-phenylalanine, and L-phenylalanine were significantly up-regulated on D15 compared to D8, while L-proline, taurine, and L-methionine increased slightly, but the difference was not significant (Figure 10). Dopamine and its precursors, tyramine, and L-tyrosine decreased significantly on D22 as compared to D15, and most amino acids were also down-regulated, including L-tyrosine, L-proline, D-ornithine, L-lysine, L-tryptophan, DL-phenylalanine, and L-kynurenine; only L-glutamine and L-pyroglutamic acid increased significantly (Figure 11). Trends in the main analyzed serum metabolites at different stages are shown in Table 5.

## 4. Discussion

Mass-spectrometry-based metabolomics has been utilized across several domains to identify biomarkers and facilitate early and prenatal diagnosis [20]. Disturbances of the metabolome that surface from physiological status changes, such as pregnancy, or as a consequence of disease are evident through distinctive metabolite patterns within tissues or bodily fluids [21]. Comprehensive measurement via metabolomics and complementary multivariate analysis identified significant metabolites as an indicator of the embryonic diapause state. This helps to facilitate further research into the underlying mechanisms.

Minks are mated from late February to late March, and the gestation period is 37 to 91 days, with an average of 47 days [22,23]. After fertilization, the zygote undergoes 5–6 cellular divisions, forming a morula that eventually develops into a blastocyst. Subsequently, nocturnal melatonin levels increase, resulting in diminished prolactin levels, followed by the entry of the blastocyst into stagnation or diapause. The blastocyst migrates to the uterine horn and enters a pre-diapause phase that lasts for 6–8 days. During this stage, the blastocyst remains non-implanted and unattached and can remain free in the uterine horns for a span of 6–31 days. Reactivation of diapause occurs when the photoperiod increases after the equinox, leading to a rise in circulating prolactin and a subsequent increase in ovarian progesterone synthesis. The blastocyst ultimately attaches to the uterine horn and develops rapidly to fetal maturity, usually around 30 days [1,3]. Therefore, we examined changes in metabolites in maternal plasma at D1 (fertilization), D8 (diapause), D15 (reactivation), and D22 and D29 (post-implantation). As we all know, the embryonic diapause that occurs in mink is regulated by the photoperiod. The short daylight (<12 h) before the vernal equinox induces an increase in the release of melatonin from the pineal gland. The secretion of prolactin is suppressed, and the production of progesterone from the corpus luteum is reduced [3,24]. Prolactin may stimulate the growth and development of the mammary glands and sustain lactation. Additionally, it is a vital regulator of embryo implantation. Dopamine is crucial in suppressing the release of endogenous prolactin [25,26]. Therefore, among the many metabolites analyzed by metabolomics, we place particular emphasis on dopamine and amino acids.

Our study discovered a significant increase in dopamine levels in D8 compared to D1, as well as a higher concentration of dopamine on D15 compared to D1. Additionally, the precursors of dopamine, L-phenylalanine, tyrosine, and tyramine were also higher on D8 and D15 compared to D1. Previous research has shown that both tyrosine and tyramine can raise dopamine levels [27,28]. Notably, L-phenylalanine acted as a synthetic precursor of dopamine [29], which also impacts mouse embryo implantation negatively by disrupting cytokine-based immunity and oxidative stress in the uterus [30]. So, we speculated that a greater amount of L-phenylalanine in the embryo during diapause stages could have detrimental consequences for both polyamine synthesis and embryo implantation. In addition, it was discovered that on D15, there was an increase in dopamine and its precursors, L-phenylalanine, tyramine, and L-tyrosine, when compared to D8. After D22, however, a decrease in these substances was observed as compared to D15.

Amino acids play a key role in female and male reproduction [31,32,33]. Many in vitro studies on mouse embryos have demonstrated that amino acids, especially essential amino acids, have a vital role in embryo development and blastocyst activation [34,35]. Essential and nonessential amino acid transport benefits pre-implantation mouse embryo development. Transport of essential amino acids is vital for viable development of the embryo, particularly after the eighth cell stage. Our study found that most amino acids increased to higher levels at D15 than at D8, especially L-kynurenine, L-lysine, and L-arginine, which are polyamine precursors. Leucine plays an important role in embryonic development [36,37,38]. The researchers demonstrated that amino acids induce trophectoderm motility and mouse embryo implantation by activating the mTOR signaling pathway. One of the most important functions of leucine is to activate the mTOR signaling pathway. It has been reported that the up-regulation of the leucine transporter SLC6A14 induced embryo activation [39], which is consistent with the results of our metabolomics analysis. The intake of mammalian animal fetal taurine is through the mother’s placenta or breastfeeding. Taurine is involved in regulating the proliferation of neural progenitor cells, the migration of newly generated neurons, and the formation of neuronal synapses during fetal development. Taurine is an intriguing molecule that links the mother with the fetus or bonding [40,41]. Ornithine is a non-essential amino acid, which plays a central role in the urea cycle. It serves as a vital component in the creation of proline, polyamines, and citrulline [24,42]. Ornithine, arginine, and proline, along with methionine, act as methyl donors in the polyamine synthesis process [43]. Research indicates that proline serves as a primary substrate for polyamine synthesis through the action of proline oxidase, ornithine aminotransferase, and ornithine decarboxylase in placentae [44,45]. At the same time, polyamines are essential for early embryonic development and successful pregnancy outcome in mammals [43,46]. Thus, increased amino acid levels may induce a state of reactivation in the embryo. A substantial reduction in most amino acids occurred after D22 and D29, suggesting that amino acids play a vital function in mink oocyte maturation and embryo development.

The changes in metabolites during pregnancy may be related to the diapause and reactivation of the embryo. Metabolomics analysis of differentially expressed metabolites, to determine the state of pregnancy, is helpful to understand the diapause of mink embryos. Our study recognizes that the metabolite expression level can be affected by the stress endured by the mink while collecting blood multiple times. Therefore, it is necessary to further verify the impact of alterations in amino acids and dopamine on pregnancy in minks through in vitro embryo culture and in vivo experiments. Subsequent experiments should explore changes in mink metabolites during pregnancy more comprehensively.

## 5. Conclusions

In this study, we compared and analyzed the serum metabolomic results during the D29 period of mating to identify metabolite fluctuations at different stages of early mink pregnancy. Our research focused on dopamine and amino acid changes, as these metabolites could be crucial for comprehending embryonic diapause, reactivation, and implantation. The study is significant for shedding light on mink embryonic diapause and determining mink pregnancy status through metabolite changes.

## Figures and Tables

**Figure 1 genes-14-01759-f001:**
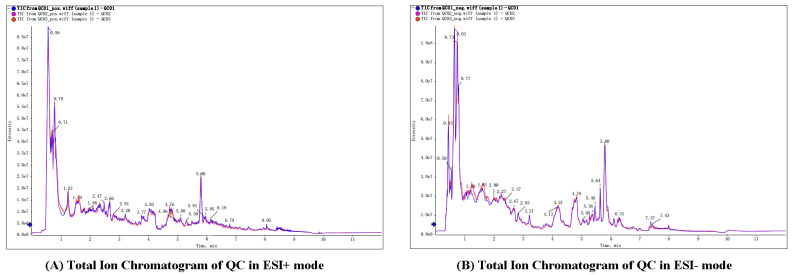
Total ion chromatogram of QC in ESI+ mode (**A**) and ESI- mode (**B**). All QC samples’ TIC peak retention time and peak area overlap very well, indicating that the instrument stability is very good.

**Figure 2 genes-14-01759-f002:**
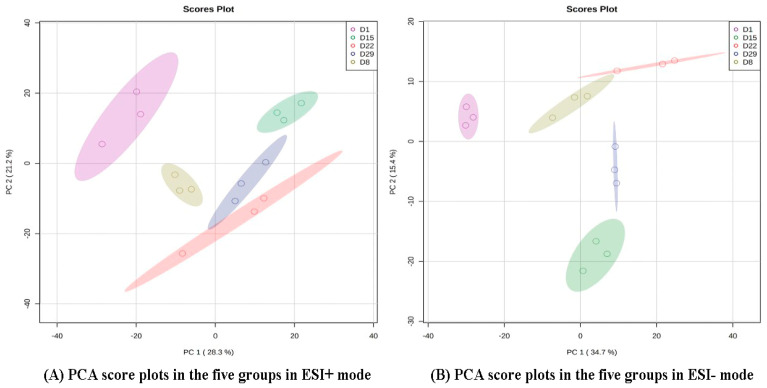
PCA score plots in the five groups in ESI+ mode (**A**) and ESI- mode (**B**). The color and shape of the symbols indicate different groups.

**Figure 3 genes-14-01759-f003:**
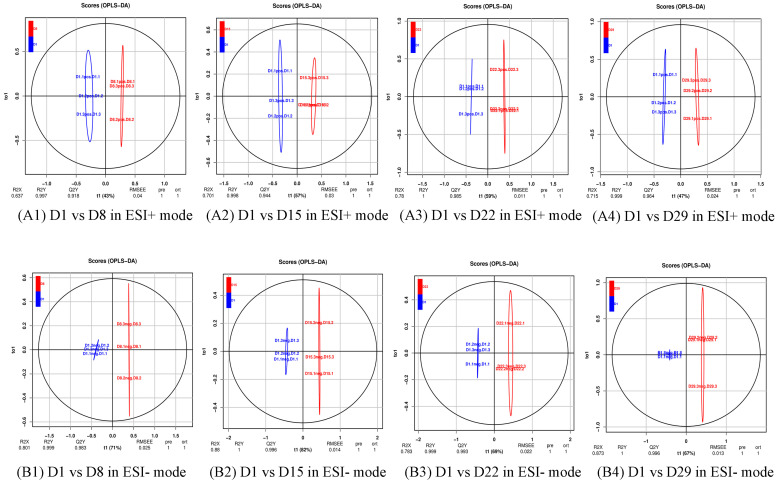
OPLS-DA score plot of the four comparison groups. R2X and R2Y respectively represent the interpretation rate of the X and Y matrices of the built model, and Q2 represents the predictive ability of the model. When Q2 > 0.5, it can be considered an effective model. Q2 > 0.9 is an excellent model. (**A1**) D1 vs. D8 in ESI+ mode, (**B1**) D1 vs. D8 in ESI- mode; (**A2**) D1 vs. D15 in ESI+ mode, (**B2**) D1 vs. D15 in ESI- mode; (**A3**) D1 vs. D22 in ESI+ mode, (**B3**) D1 vs. D22 in ESI- mode; (**A4**) D1 vs. D29 in ESI+ mode, (**B4**) D1 vs. D29 in ESI- mode.

**Figure 4 genes-14-01759-f004:**
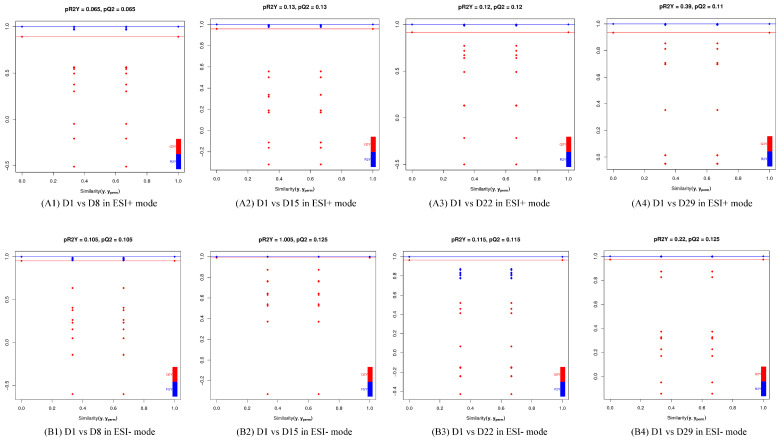
Validation plot of the OPLS-DA model. The horizontal line corresponded to R2 and Q2 of the original model, and the blue point and red point represented R2′ and Q2′ of the model after Y replacement, respectively. (**A1**) D1 vs. D8 in ESI+ mode, (**B1**) D1 vs. D8 in ESI- mode; (**A2**) D1 vs. D15 in ESI+ mode, (**B2**) D1 vs. D15 in ESI- mode; (**A3**) D1 vs. D22 in ESI+ mode, (**B3**) D1 vs. D22 in ESI- mode; (**A4**) D1 vs. D29 in ESI+ mode, (**B4**) D1 vs. D29 in ESI- mode.

**Figure 5 genes-14-01759-f005:**
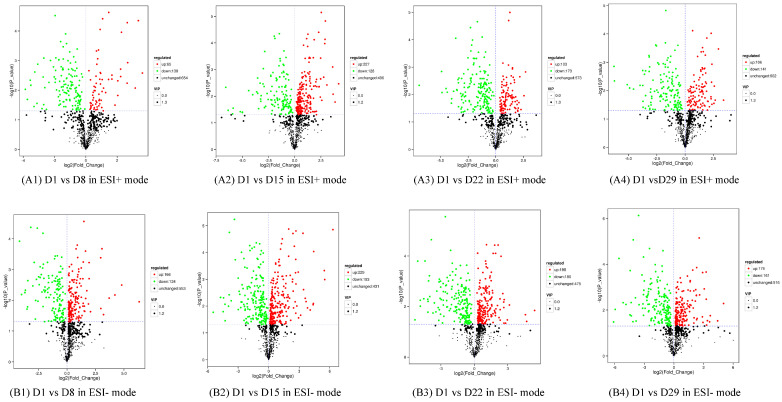
Volcano plot with differential metabolites of the four comparison groups. Each point in the figure represents a metabolite, the abscissa represents the log2-fold change of the group compared to the substance, and the ordinate represents the log10-*p*-value of the Student’s t-test. The scatter size represents the VIP value of the OPLS-DA model. Green dots represent significantly down-regulated metabolites, red dots represent significantly up-regulated metabolites, and black dots represent metabolites detected but not significantly different. (**A1**) D1 vs. D8 in ESI+ mode, (**B1**) D1 vs. D8 in ESI- mode; (**A2**) D1 vs. D15 in ESI+ mode, (**B2**) D1 vs. D15 in ESI- mode; (**A3**) D1 vs. D22 in ESI+ mode, (**B3**) D1 vs. D22 in ESI- mode; (**A4**) D1 vs. D29 in ESI+ mode, (**B4**) D1 vs. D29 in ESI- mode.

**Figure 6 genes-14-01759-f006:**
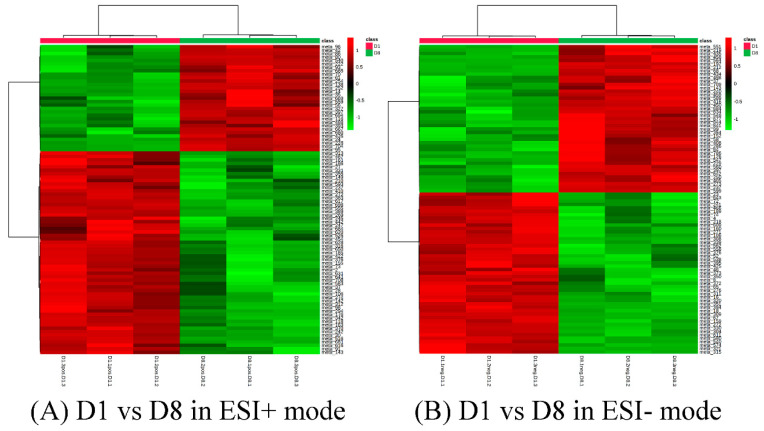
Cluster analysis of D1 vs. D8 in ESI+ mode (**A**) and ESI- mode (**B**). Each row represents a different grouping, and each column represents a different metabolite. The color indicates the difference multiple of metabolites, and the darker the color, the greater the multiple. Red means up-regulation, and green means down-regulation.

**Figure 7 genes-14-01759-f007:**
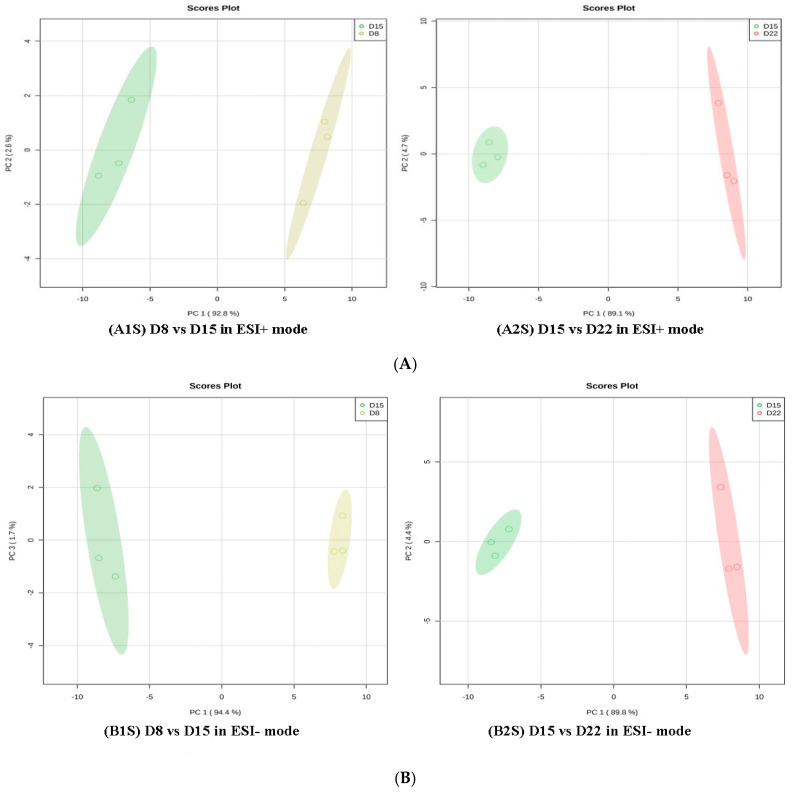
PCA score plots in D8 vs. D15 and D15 vs. D22 in ESI+ mode (**A**) and ESI- mode (**B**). (**A1S**) D8 vs. D15 in ESI+ mode, (**B1S**) D8 vs. D15 in ESI- mode; (**A2S**) D15 vs. D22in ESI+ mode, (**B2S**) D15 vs. D22 in ESI- mode.

**Figure 8 genes-14-01759-f008:**
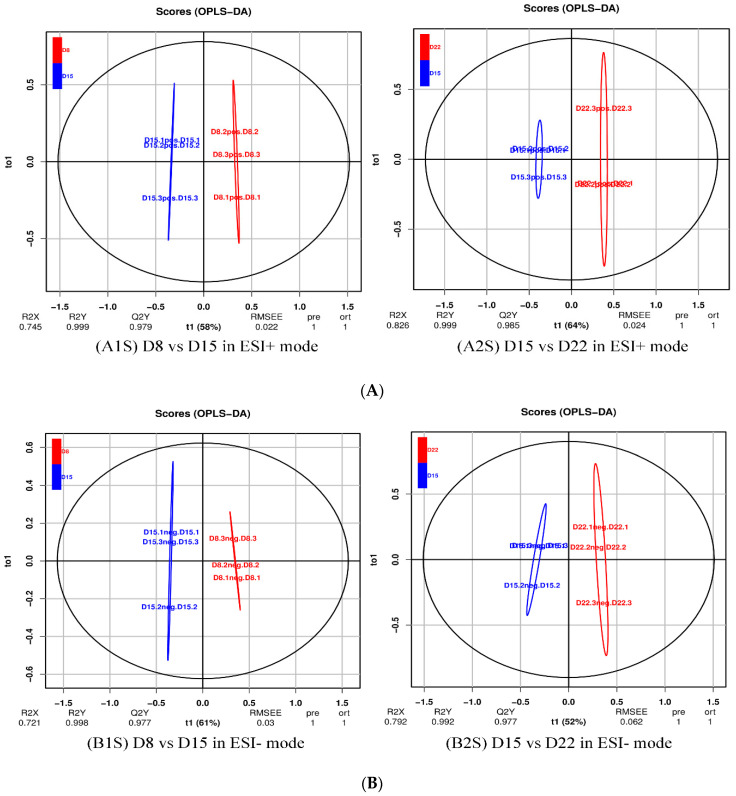
OPLS-DA score plot of D8 vs. D15 and D15 vs. D22 in ESI+ mode (**A**) and ESI- mode (**B**). (**A1S**) D8 vs. D15 in ESI+ mode, (**B1S**) D8 vs. D15 in ESI- mode; (**A2S**) D15 vs. D22 in ESI+ mode, (**B2S**) D15 vs. D22 in ESI- mode.

**Figure 9 genes-14-01759-f009:**
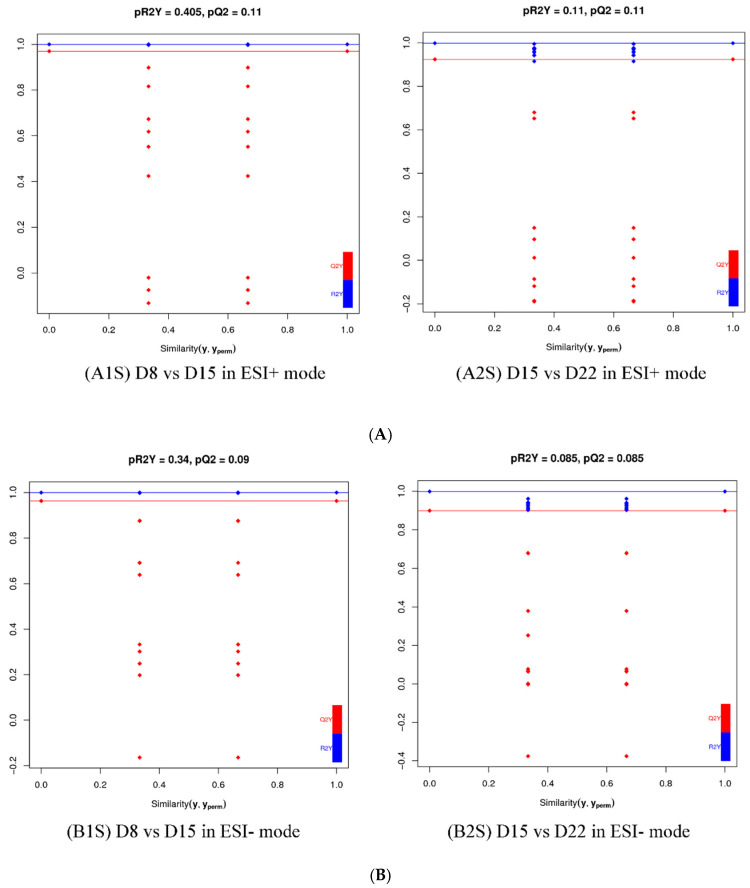
Validation plot of OPLS-DA model D8 vs. D15 and D15 vs. D22in ESI+ mode (**A**) and ESI- mode (**B**). The horizontal line corresponds to R2 and Q2 of the original model, and the blue point and red point represent R2′ and Q2′ of the model after Y replacement, respectively. (**A1S**) D8 vs. D15 in ESI+ mode, (**B1S**) D8 vs. D15 in ESI- mode; (**A2S**) D15 vs. D22 in ESI+ mode, (**B2S**) D15 vs. D22 in ESI- mode.

**Figure 10 genes-14-01759-f010:**
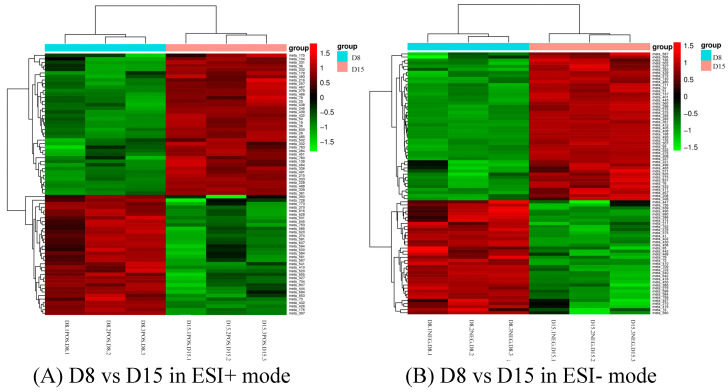
Cluster analysis of D8 vs. D15 in ESI+ mode (**A**) and ESI- mode (**B**). Each row represents a different grouping, and each column represents a different metabolite. The color indicates the difference multiple of metabolites, and the darker the color, the greater the multiple. Red means up-regulation, and green means down-regulation.

**Figure 11 genes-14-01759-f011:**
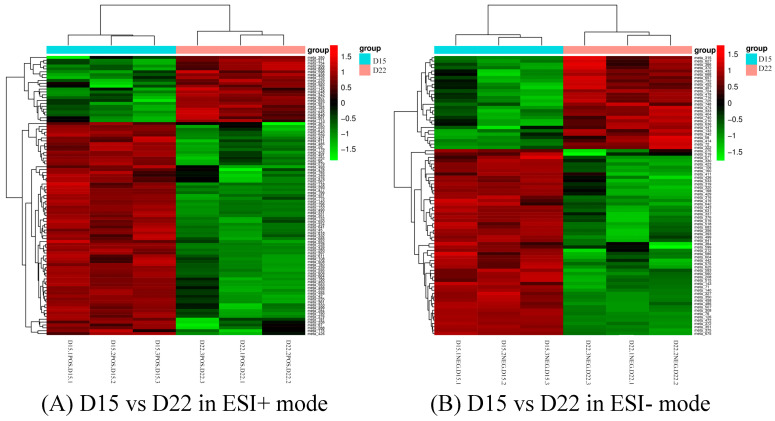
Cluster analysis of D15 vs. D22 in ESI+ mode (**A**) and ESI- mode (**B**). Each row represents a different metabolite, and each column represents a different grouping. The color indicates the difference multiple of metabolites, and the darker the color, the greater the multiple. Red means up-regulation, and green means down-regulation.

**Table 1 genes-14-01759-t001:** Amino acids and dopamine change (D1 vs. D8).

No.	Name	*p*-Value	VIP	Regulated	KEGG Pathway Annotation
1	L-proline	0.003	1.300	down	Metabolic pathways (ko01100); arginine and proline metabolism (ko00330)
2	L-valine	0.004	1.294	down	Metabolic pathways (ko01100); biosynthesis of amino acids (ko01230)
3	L-threonine	0.015	1.225	down	Valine, leucine, and isoleucine biosynthesis (ko00290)
4	Taurine	0.049	1.201	down	Taurine and hypotaurine metabolism (ko00430)
5	L-leucine	0.021	1.204	down	Metabolic pathways (ko01100); biosynthesis of amino acids (ko01230)
6	L-methionine	0.036	1.249	up	Cysteine and methionine metabolism (ko00270)
7	L-phenylalanine	0.001	1.358	up	Phenylalanine, tyrosine, and tryptophan biosynthesis (ko00400); phenylalanine metabolism (ko00360)
8	L-glutamine	0.030	1.198	down	Taurine and hypotaurine metabolism (ko00430); arginine and proline metabolism (ko00330)
9	Tyramine	0.022	1.474	up	Tyrosine metabolism (ko00350)
10	Dopamine	0.000	1.543	up	Metabolic pathways (ko01100); prolactin signaling pathway (ko04917)
11	D-ornithine	0.004	1.487	down	D-arginine and D-ornithine metabolism (ko00472); metabolic pathways (ko01100)
12	L-tyrosine	0.000	1.535	up	Phenylalanine metabolism (ko00360); dopaminergic synapse (ko04728); prolactin signaling pathway (ko04917)
13	L-lysine	0.002	1.518	down	Lysine degradation (ko00310); metabolic pathways (ko01100)
14	L-kynurenine	0.000	1.533	down	Tryptophan metabolism (ko00380)

**Table 2 genes-14-01759-t002:** Amino acids and dopamine change (D1 vs. D15).

No.	Name	*p*-Value	VIP	Regulated	KEGG Pathway Annotation
1	L-glutamate	0.008	1.190	down	Metabolic pathways (ko01100); biosynthesis of amino acids (ko01230)
2	L-methionine	0.004	1.258	up	Biosynthesis of amino acids (ko01230); cysteine and methionine metabolism (ko00270)
3	L-proline	0.007	1.260	down	Arginine and proline metabolism (ko00330); Metabolic pathways (ko01100)
4	L-threonine	0.015	1.212	down	Glycine, serine, and threonine metabolism (ko00260); metabolic pathways (ko01100)
5	L-phenylalanine	0.004	1.270	up	Biosynthesis of amino acids (ko01230); phenylalanine metabolism (ko00360)
6	L-tryptophan	0.024	1.218	down	Phenylalanine, tyrosine, and tryptophan biosynthesis (ko00400)
7	L-leucine	0.024	1.182	down	Valine, leucine, and isoleucine degradation (ko00280)
8	L-valine	0.011	1.245	down	Valine, leucine, and isoleucine biosynthesis (ko00290)
9	L-lysine	0.041	1.063	down	Lysine biosynthesis (ko00300); metabolic pathways (ko01100)
10	Taurine	0.047	1.252	down	Metabolic pathways (ko01100); taurine and hypotaurine metabolism (ko00430)
11	L-lysine	0.011	1.338	down	Lysine biosynthesis (ko00300); metabolic pathways (ko01100)
12	L-arginine	0.006	1.383	down	Biosynthesis of amino acids (ko01230); metabolic pathways (ko01100)
13	L-glutamate	0.012	1.340	down	Alanine, aspartate, and glutamate metabolism (ko00250)

**Table 3 genes-14-01759-t003:** Amino acids and dopamine change (D1 vs. D22).

No.	Name	*p*-Value	VIP	Regulated	KEGG Pathway Annotation
1	L-valine	0.006	1.207	down	Valine, leucine, and isoleucine degradation (ko00280)
2	L-glutamine	0.008	1.200	down	Biosynthesis of amino acids (ko01230); metabolic pathways (ko01100)
3	L-phenylalanine	0.001	1.270	up	Phenylalanine metabolism (ko00360); biosynthesis of amino acids (ko01230)
4	L-tryptophan	0.022	1.132	down	Phenylalanine, tyrosine, and tryptophan biosynthesis (ko00400)
5	L-threonine	0.011	1.196	down	Biosynthesis of amino acids (ko01230); metabolic pathways (ko01100)
6	L-methionine	0.008	1.241	up	Metabolic pathways (ko01100); cysteine and methionine metabolism (ko00270)
7	L-proline	0.002	1.256	down	Arginine and proline metabolism (ko00330); metabolic pathways (ko01100)
8	L-leucine	0.005	1.217	down	Biosynthesis of amino acids (ko01230); metabolic pathways (ko01100)
9	Dopamine	0.000	1.543	up	Metabolic pathways (ko01100); prolactin signaling pathway (ko04917)
10	D-ornithine	0.004	1.487	down	D-arginine and D-ornithine metabolism (ko00472); metabolic pathways (ko01100)
11	L-tyrosine	0.000	1.535	up	Phenylalanine metabolism (ko00360); dopaminergic synapse (ko04728); prolactin signaling pathway (ko04917)
12	L-lysine	0.002	1.518	down	Lysine degradation (ko00310); metabolic pathways (ko01100)

**Table 4 genes-14-01759-t004:** Amino acids and dopamine change (D1 vs. D29).

No.	Name	*p*-Value	VIP	Regulated	KEGG Pathway Annotation
1	L-phenylalanine	0.046	1.200	up	Phenylalanine, tyrosine, and tryptophan biosynthesis (ko00400)
2	L-glutamate	0.016	1.190	down	Arginine biosynthesis (ko00220)
3	L-tryptophan	0.031	1.288	down	Phenylalanine, tyrosine, and tryptophan biosynthesis (ko00400)
4	L-phenylalanine	0.022	1.291	up	Phenylalanine, tyrosine, and tryptophan biosynthesis (ko00400)
5	L-alanine	0.025	1.340	down	Metabolic pathways (ko01100); cysteine and methionine metabolism (ko00270)
6	L-glutamate	0.048	1.254	down	Arginine and proline metabolism (ko00330); metabolic pathways (ko01100)
7	L-lysine	0.031	1.339	down	Biosynthesis of amino acids (ko01230); metabolic pathways (ko01100)

**Table 5 genes-14-01759-t005:** Dynamics of the serum levels of the main analyzed compounds.

No.	Name	D1 vs. D8	D8 vs. D15	D15 vs. D22	D22 vs. D29
1	Dopamine	up	up	down	down
2	Tyramine	up	up	down	down
3	L-phenylalanine	up	up	down	down
4	L-tyrosine	up	up	down	down
5	Tyrosine	up	up	down	down
6	L-kynurenine	down	up	down	down
7	L-lysine	down	up	down	down
8	L-arginine	down	up	down	down
9	D-ornithine	down	down	down	down
10	Leucine	down	up	down	down

## Data Availability

All data involved in this article are original and available from the corresponding authors on reasonable request.

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
