# Peer review of "Metabolomic Profiling of Female Mink Serum during Early to Mid-Pregnancy to Reveal Metabolite Changes"

_genes, 2023, doi:10.3390/genes14091759_

Round 1
Reviewer 1 Report (Previous Reviewer 2)
The authors took into account all previous comments and significantly improved MS. I have no major points to the article.
Minor point
The authors speculate that high levels of L-phenylalanine may have a negative effect on embryo implantation (lines 341-343). How can the authors explain the increase in the level of L-phenylalanine at D15, which they interpret as a stage of reactivation?
Author Response
L-phenylalanine is a synthetic precursor of dopamine, which has been shown to negatively affect embryo implantation by disrupting cytokine immunity and oxidative stress in utero. At D8, the embryos were in the diapause phase, a stage in which we found high levels of L-phenylalanine. Therefore, we speculated that more L-phenylalanine in the embryo during the diapause phase may adversely affect polyamine synthesis and embryo implantation. While at D15, L-phenylalanine was increased, alongside all other essential amino acids were also increased compared to D8. Such as Arginine , Lysine , Ornithine and Leucine. Amino acids, especially essential amino acids, play an important role in embryonic development and blastocyst activation, so we considered D15 as a reactivation stage to prepare for embryo bedding. Our study mentioned that it will focus on dopamine as well as amino acid changes. So the paragraph focuses on the changes in dopamine at various stages. The relevant revised in lines 317-328.

Reviewer 2 Report (New Reviewer)
Genes-2584199-Peer-Review Report-v1
The authors centred their studies on the changes in metabolites in the early
pregnancy of mink by comparing and analysing serum metabolites up to twenty-nine
days after mating. The study results were adequately discussed, citing relevant and
recently peer-reviewed works. This work will significantly add to knowledge in this
area.
However, I observed some areas that should be checked and corrected
accordingly.
1. The authors stated that each of the female minks was mated with a male,
according to the normal practice. Did they carry out fertility tests prior to the
mating? Normal practice, what is it? Write a sentence or two about this.
2. The authors stated that the minks were fed sufficient food. What is the nature
of this food? This must be stated in a sentence or two.
3. Under Data processing and pattern recognition, at what level is P significant?
It should be clearly stated; do not assume that your readers know.
4. Table 5 is presented out of place; it should be presented under results rather
than after the discussion.
5. There is no need for a table of abbreviations used in the text. These
abbreviations have already been defined at first mention in the text, except for
Embryonic Stem Cells (ESC), which is not part of the text (check). This is not a
thesis, and this table is needless. It should be discarded.
6. There is no conclusion of this study. Even though it is not mandatory, its
inclusion will add value to the work. Some readers will read only the abstract
and conclusions. The authors may consider adding it.
Furthermore, the authors should check and correct the following:
a) Line 15 For clarity, remove the phrase, ‘and then’.
b) Line 18 For clarity, delete the phrase, ‘ion mode’.
c) Line 26 Add the article ‘the’ at the beginning of the sentence…The birth…..
d) Line 28 The spelling of ‘favorable’ is a non-British variant. For consistency,
please change it to British English spelling. Check others like ‘color’ etc, used in
the manuscript.
e) Line 30,31: Change the wording. Suggestion: From the vernal equinox, the
photoperiod increases gradually, stimulating prolactin secretion from the
pituitary gland [2, 3].
f) Line 72 There is inconsistent hyphenation with the word ‘pre-implantation’.
Check and use one throughout the text.
g) Line 88 Remove the preposition ‘during’ before implantation.
h) Line 133 check source temperature ‘650oC’
i) Line 183 Incorrect article usage, ‘a’ should be used instead of
‘the’. Suggestion: …. model had a high predictive…
j) References Check reference numbers 18 and 28.
This is included in the attached file.
Author Response
The authors centred their studies on the changes in metabolites in the early pregnancy of mink by comparing and analysing serum metabolites up to twenty-nine days after mating. The study results were adequately discussed, citing relevant and recently peer-reviewed works. This work will significantly add to knowledge in this area.However, I observed some areas that should be checked and corrected accordingly.
Point 1: The authors stated that each of the female minks was mated with a male, according to the normal practice. Did they carry out fertility tests prior to the mating? Normal practice, what is it? Write a sentence or two about this.
Response 1: Of the minks that we selected, all were normal and fertile. “Normal practice” means that in order to control the variables and ensure the accuracy of the experiment to a greater extent, we only chose a fertile male mink and used artificial insemination for mating (lines 105-106).
Point 2: The authors stated that the minks were fed sufficient food. What is the nature of this food? This must be stated in a sentence or two.
Response 2: We use specialised feeds for mink, including animal feeds, plant feeds and supplementary feeds (lines 104-105).
Point 3: Under Data processing and pattern recognition, at what level is P significant? It should be clearly stated; do not assume that your readers know.
Response 3: The differential metabolites were selected by the combination of the P-value of the ANOVA and variable importance in the projection (VIP) values of the OPLS-DA model. The results are considered statistically significant as the P-value was less than 0.05 and the VIP value was larger than 1. The log2-fold change was used to show how these selected differential metabolites varied between groups (lines 157-161).
Point 4: Table 5 is presented out of place; it should be presented under results rather than after the discussion.
Response 4: Table 5 has been moved to line 278. In the Results section, we analysed the changes in serum metabolites under different stages. The main metabolite trends are available in Table 5. In addition, the main metabolites in the table are also discussed in the Discussion section.
Point 5: There is no need for a table of abbreviations used in the text. These abbreviations have already been defined at first mention in the text, except for Embryonic Stem Cells (ESC), which is not part of the text (check). This is not a thesis, and this table is needless. It should be discarded.
Response 5: The table of abbreviations has been removed. And upon inspection, Embryonic Stem Cells (ESC) are indeed not present in this paper.
Point 6: There is no conclusion of this study. Even though it is not mandatory, its inclusion will add value to the work. Some readers will read only the abstract and conclusions. The authors may consider adding it. Furthermore, the authors should check and correct the following:
- a) Line 15 For clarity, remove the phrase, ‘and then’.
- b) Line 18 For clarity, delete the phrase, ‘ion mode’.
- c) Line 26 Add the article ‘the’ at the beginning of the sentence…The birth…..
- d) Line 28 The spelling of ‘favorable’ is a non-British variant. For consistency, please change it to British English spelling. Check others like ‘color’ etc, used in the manuscript.
- e) Line 30,31: Change the wording. Suggestion: From the vernal equinox, the photoperiod increases gradually, stimulating prolactin secretion from the pituitary gland [2, 3].
- f) Line 72 There is inconsistent hyphenation with the word ‘pre-implantation’. Check and use one throughout the text.
- g) Line 88 Remove the preposition ‘during’ before implantation.
- h) Line 133 check source temperature ‘650oC’
- i) Line 183 Incorrect article usage, ‘a’ should be used instead of ‘the’. Suggestion: …. model had a high predictive…
- j) References Check reference numbers 18 and 28.
Response 6: Conclusion added at line 364.
(In this study, we compared and analyzed the serum metabolomic results during D29 period of mating to identify metabolite fluctuations at different stages of early mink pregnancy. Our research focused on dopamine and amino acid changes, as these metabolites could be crucial for comprehending embryonic diapause, reactivation, and implantation. The study is significant for shedding light on mink embryonic diapause and determining mink pregnancy status through metabolite changes.)
- a) Line 19 deleted “and then”
- b) Line 22 deleted “ion mode”
- c) Line 31 added “the” …The birth…
- d) Line 33 Replaced fovorable with favourable. Replaced color with colour in full text.
- e) Line 35-36 From the vernal equinox, the photoperiod increases gradually, stimulating prolactin secretion from the pituitary gland [2, 3].
- f) Checked and revised in full.
- g) Line 93 deleted “during”
- h) Line 138 650°C
- i) Line 186 …. model had a high predictive…
- j) Checked and revised in full.
Reviewer 3 Report (New Reviewer)
Manuscript ID: genes-2584199
Title: Metabolomic profiling of female mink serum during early to mid-pregnancy to reveal metabolite changes
The manuscript is well-written and addresses an interesting topic.
However, minor considerations must be taken in to account.
. Review the legend of figure 8, replace "Each row represents a different grouping, and each column represents a different metabolite" with "Each row represents a different metabolite, and each column represents a different grouping"
. The discussion should not include tables. Please, move and comment table 5 to the results section.
Author Response
Point 1: Review the legend of figure 8, replace "Each row represents a different grouping, and each column represents a different metabolite" with "Each row represents a different metabolite, and each column represents a different grouping"
Response 1: Complete the changes at lines 284-285.
Point 2: The discussion should not include tables. Please, move and comment table 5 to the results section.
Response 2: Table 5 has been moved to line 277 . In the results section, we summarised trends in the main serum metabolites analysed at different stages.This information can be found in Table 5. Additionally, we also have conducted a relevant summary and discussion of the results in the discussion section.

This manuscript is a resubmission of an earlier submission. The following is a list of the peer review reports and author responses from that submission.
Round 1
Reviewer 1 Report
Putrescine (1,4-diaminobutane) is the simplest of the mammalian polyamines and thought to play a role in regulation of anabolic events such as synthesis of DNA, RNA, and protein. However, it is necessary to explain the relationship between polyamine and putrescine in line 42-46.
A research necessity or interest is an acknowledged study. Was there a targeting metabolite in the metabolic profiling analysis?
Please add the approval number of the animal use for this experiment in line 73-74.
Does this mean that only 15 of the 20 minks got pregnant and only 6 of them were serum isolated? How many of each serum were you able to obtain Is there a reason why you used two serums mixed together?
Where is B1-B4 in figure 5 in line 192~202?
In line 307-308, authors assume that the higher levels of l-phenylalanine have negative effects on polyamine synthesis and embryo implantation. Why do authors claim that? There are no evidence of negative or harmful results on Day8 and Day15.
Comparison of metabolites with changes shows only D1 and D8, D1 and D15, and D1 and D22, but not D1 and D29. Any reason?
Is it possible to derive results about the trend change of the relative amount metabolites from D1, D8, D15, D22 and D29?
no special comments
Author Response
Response to Reviewer 1 Comments
Dear editor and reviewers:
Thank you for your decision and constructive comments on my manuscript. Those comments are all valuable and very helpful for revising and improving our manuscript. We have carefully considered your suggestions and have tried our best to improve and made some changes in the manuscript.The red part that has been revised according to your comments. Revision notes, point-to-point, are given as follows.
Yours sincerely,
Chunjin Li, Ph. D.
Professor of Animal Reproduction
College of Animal Sciences
Jilin University
5333 Xian Road
Changchun, Jilin 130062
P. R. China
Phone:86-431-87835142
Point 1: Putrescine (1,4-diaminobutane) is the simplest of the mammalian polyamines and thought to play a role in regulation of anabolic events such as synthesis of DNA, RNA, and protein. However, it is necessary to explain the relationship between polyamine and putrescine in line 42-46.
Response 1: The relationship between putrescine and polyamines and their important roles are described in lines 49-52. The modifications are as follows: Putrescine is the first polyamine molecule in the de novo polyamine synthesis pathway in all higher mammalian cells, and plays an important role in promoting cell division and differentiation, synthesis of macromolecules such as DNA, RNA, and proteins, and repair of inflammatory processes.
Point 2: A research necessity or interest is an acknowledged study. Was there a targeting metabolite in the metabolic profiling analysis?
Response 2: The embryonic diapause that occurs in mink is regulated by photoperiod. The short daylight (<12 h) before the vernal equinox induces an increase in the release of melatonin from the pineal gland. The secretion of prolactin is suppressed and the production of progesterone from the corpus luteum is reduced. Prolactin can promote the development and growth of mammary glands, and induce and maintain lactation. It is also an important hormone that regulates embryo implantation. And dopamine plays a key role in inhibiting the secretion of endogenous prolactin. Therefore, among the many metabolites analyzed by metabolomics, we pay special attention to dopamine and amino acids.Mink embryos develop into blastocysts and then enter a dormant phase, the reactivation of which is mainly caused by an increase in polyamines. The effect of polyamines on lagging mink embryos has been confirmed. So we took polyamine as the main target metabolite. In the present study, through metabolomic analysis, we determined the changes in dopamine, a representative substance of polyamines, as well as precursors of dopamine synthesis, such as tyrosine, L-tyrosine, and L-phenylalanine, at different times.
Point 3: Please add the approval number of the animal use for this experiment in line 73-74.
Response 3: The approval number for the use of animals in this experiment has been added in line 95. Modify as follows: All experiments on minks were guided by and with the approval of the Animal Protection and Utilization Committee (SY202003300).
Point 4:Does this mean that only 15 of the 20 minks got pregnant and only 6 of them were serum isolated? How many of each serum were you able to obtain Is there a reason why you used two serums mixed together?
Response 4: Of the 20 mink we selected, only 15 became pregnant and gave birth. Follow-up of subsequent pregnancies, we selected six mink serum samples from the same gestation period for metabolomic analysis. The total amount of blood collected for each sample was 1 mL, and approximately 400 µL of serum was obtained. Of the six serum samples from the same sampling date, we mixed two mink sera with the same genetic background together as three biological replicates. Related changes in lines 106-109.
Point 5: Where is B1-B4 in figure 5 in line 192~202?
Response 5: Figure 5 has been completed at line 206.
Point 6: In line 307-308, authors assume that the higher levels of l-phenylalanine have negative effects on polyamine synthesis and embryo implantation. Why do authors claim that? There are no evidence of negative or harmful results on Day8 and Day15.
Response 6: Based on the comparative analysis of metabolite changes, we found that the dopamine content was significantly increased at D8 and D15 compared to D1. In addition, the precursors of dopamine, L-phenylalanine, tyrosine and tyramine, were higher at D8 and D15 than at D1, and L-phenylalanine, as a precursor of dopamine synthesis, has been shown to negatively affect embryo implantation by disrupting cytokine immunity and oxidative stress in the uterus. Therefore, we suggest that higher levels of L-phenylalanine may be one of the reasons why embryonic stunting at D8 and D15. In addition, we found that dopamine and its precursors, L-phenylalanine, tyramine, and L-tyrosine, were increased on D15 as compared to D8, were lower on D22 than on D15.So we speculate that higher level of L-phenylalanine in the embryo of diapause stages D15 may have negative effects on polyamine synthesis and embryo implantation.
Changes in the manuscript are located in lines 315-321: In addition, we also found that dopamine and its precursors, L-phenylalanine, tyramine, and L-tyrosine, were increased on D15 as compared to D8, ​were lower on D22 than on D15. L-phenylalanine is a synthetic precursor of dopamine, which also impacts mouse embryo implantation negatively by disrupting cytokine-based immunity and oxidative stress in the uterus.So we speculate that higher level of L-phenylalanine in the embryo of diapause stages D15 may have negative effects on polyamine synthesis and embryo implantation.
Point 7: Comparison of metabolites with changes shows only D1 and D8, D1 and D15, and D1 and D22, but not D1 and D29. Any reason?
Response 7: In line 243, we compare D1 and D29(Table 4). At D29, we found a decreasing trend in most amino acids when compared to D1.
Point 8: Is it possible to derive results about the trend change of the relative amount metabolites from D1, D8, D15, D22 and D29?
Response 8: We compared the relative amounts of metabolites of D8,D15,D22 and D29 all with D1, revealing the changes in the relative amounts of metabolites at different times. Since many amino acids contribute to embryonic development and blastocyst activation, we focused on showing the changes of amino acids and dopamine in different groups. Trends in the major metabolites can be obtained from D8, D15, D22 and D29. Dynamics of the serum levels of the main analyzed compounds in Table 5 (line 347).
For example, L-proline, L-threonine, taurine, L-leucine, D-ornithine, L-lysine, L-valine, L-kynurenine, and L-glutamine were decreased in D8 compared to D1. In contrast, L-methionine, L-phenylalanine, and L-tyrosine were upregulated in D8. Similarly, dopamine and its precursor tyramine were significantly increased in D8 compared to D1. Dopamine was again significantly higher in D15 than in D8, and the precursors of dopamine, L-phenylalanine ,tyrosine and L-tyrosine, were also higher in D15 than in D8, but L-glutamine and D-ornithine were decreased. Most amino acids increased to higher levels at D15 compared to D8. In particular, L-kynurenine, L-lysine and L-arginine as polyamine precursors, as well as DL-phenylalanine and L-phenylalanine were significantly up-regulated at D15, whereas L-proline, taurine, and L-methionine increased slightly, but the difference was not significant. After D22, most amino acids decreased significantly.Dopamine and its precursors such as tyrosine and L-tyrosine were significantly decreased at D22 compared to D15, and most amino acids were also down-regulated, including L-tyrosine, L-proline, D-ornithine, L-lysine, L-tryptophan, DL-phenylalanine, and L-kynurenine; only L-glutamine and L-pyroglutamic acid were significantly increased. At D29, the amino acid species and trends were consistent with D22.

Reviewer 2 Report
In the submitted MS, the authors present the results of a metabolomic analysis of mink serum at the different stages of pregnancy. Using ultra-performance liquid chromatography/mass spectrometry, they
detected significant changes in the levels of several amino acids and their derivatives, including dopamine, as pregnancy progressed. The authors discuss the results obtained mainly in the context of the phenomenon of embryonic diapause and reactivation of the embryo. Obviously, studies aimed at deciphering the mechanisms of early development, especially using high-precision analytical methods, are relevant, and the results obtained are of potential interest to a wide range of specialists. However, the MS requires significant improvement.
Major points
1. The authors should clearly relate the chronology of the development of the mink embryo and the date of sampling, perhaps even in a graphical form. In the present version of the MS, nowhere is explained which stage of pregnancy (pre-diapause, diapause, reactivation, post-implantation period) corresponds to D8, D15, etc. This relation between the stage of ontogeny and the experimental point should also be traced in the Discussion section.
2. The authors analyze the chemical composition of blood serum and discuss their results in the context of the effect of certain compounds on developing embryos. At the same time, the authors compare the stages of ontogeny with varying degrees of the relationship between the fetus and mother. At the D1 stage, the zygote is in the oviduct; at the D8 stage, the blastocyst is in the uterine cavity; later on, the
blastocyst is implanted in the uterine wall and the placental circulation system is formed. Thus, in the early stages studied, the environment of the embryo is created by the epithelial cells of the oviducts and uterus and may differ from the blood serum. For example, the glucose level in the oviduct cavity is lower than in the blood serum. The authors should present arguments that changes in the content of the main discussed compounds in the serum reflect changes in their content in the cavity of the oviduct and uterus.
3. Authors should provide summary graphs or charts showing the dynamics of the serum levels of the main analyzed compounds.
Minor points
1. The design of the figures could be more informative and user friendly. For example, instead of “Fig. 3-A1”, better “D1 vs D8 in ESI+ mode”, etc.
2. Some files in supplementary materials duplicate the figures in the MS.
3. Line 21: Neovison vison should be italicized (it is a species name)
4. Line 22: The incorrect statement “…the embryo is discharged from the uterus…” Embryos in a state of diapause are located in the uterine cavity.
5. Line 34: The incorrect statement “…there is almost no mitosis or protein synthesis…” Indeed, the metabolic activity of embryos in the state of diapause is reduced; however, a basic level of protein synthesis is required to maintain the viability of embryos.
6. Line 74: The characteristics of the experimental animals should be specified (age, body weight).
7. Line 205: The incorrect statement “Most of the differentially-expressed amino acids…” The term “differential expression” applies only to genes.
Author Response
Response to Reviewer 2 Comments
Dear editor and reviewers:
Thank you for your decision and constructive comments on my manuscript. Those comments are all valuable and very helpful for revising and improving our manuscript. We have carefully considered your suggestions and have tried our best to improve and made some changes in the manuscript.The red part that has been revised according to your comments. Revision notes, point-to-point, are given as follows. We look forward to hearing from you soon.
Yours sincerely,
Chunjin Li, Ph. D.
Professor of Animal Reproduction
College of Animal Sciences
Jilin University
5333 Xian Road
Changchun, Jilin 130062
P. R. China
Phone:86-431-87835142
In the submitted MS, the authors present the results of a metabolomic analysis of mink serum at the different stages of pregnancy. Using ultra-performance liquid chromatography/mass spectrometry, they
detected significant changes in the levels of several amino acids and their derivatives, including dopamine, as pregnancy progressed. The authors discuss the results obtained mainly in the context of the phenomenon of embryonic diapause and reactivation of the embryo. Obviously, studies aimed at deciphering the mechanisms of early development, especially using high-precision analytical methods, are relevant, and the results obtained are of potential interest to a wide range of specialists. However, the MS requires significant improvement.
Point 1: The authors should clearly relate the chronology of the development of the mink embryo and the date of sampling, perhaps even in a graphical form. In the present version of the MS, nowhere is explained which stage of pregnancy (pre-diapause, diapause, reactivation, post-implantation period) corresponds to D8, D15, etc. This relation between the stage of ontogeny and the experimental point should also be traced in the Discussion section.
Response 1: Thanks for your advices, relevant discussion on the lines 292-303.
Point 2: The authors analyze the chemical composition of blood serum and discuss their results in the context of the effect of certain compounds on developing embryos. At the same time, the authors compare the stages of ontogeny with varying degrees of the relationship between the fetus and mother. At the D1 stage, the zygote is in the oviduct; at the D8 stage, the blastocyst is in the uterine cavity; later on, the blastocyst is implanted in the uterine wall and the placental circulation system is formed. Thus, in the early stages studied, the environment of the embryo is created by the epithelial cells of the oviducts and uterus and may differ from the blood serum. For example, the glucose level in the oviduct cavity is lower than in the blood serum. The authors should present arguments that changes in the content of the main discussed compounds in the serum reflect changes in their content in the cavity of the oviduct and uterus.
Response 2: Embryo implantation is a complex process which involves biochemical and physiological interactions between an implantation-competent blastocyst and a receptive uterus. Across mammals, early embryonic development is supported by uterine secretions taken up through the yolk sac and other foetal membranes (histotrophic nutrition). It was found that histones in uterine secretions may help regulate preimplantation blastocyst activation as well as follicle penetration of the uterine epithelium during embryo implantation. A preimplantation embryo exists independent of blood supply, and relies on energy sources from its in vivo environment (e.g., oviduct and uterine fluid) to sustain its development. The embryos can survive in this aqueous environment because it contains amino acids, proteins, lactate, pyruvate, oxygen, glucose, antioxidants, ions, growth factors, hormones, and phospholipids,etc. Gas chromatography-mass spectrometry was used to analyze the metabolite profiles of the uterus, uterine fluid, and maternal plasma at pre-implantation and implantation.Study finds that between pre-implantation day 1 and day 4, dramatic metabolic changes were observed in the uterine fluid that could be important for blastocyst development and protection against the harsh uterine environment. In addition, metabolites in maternal plasma showed similar changes. Such as cysteine, who improves blastocyst development and viability. These results reveal how the maternal in vivo environments are biochemically modified to fulfill the dynamic demand of the pre-implantation embryo and receptive endometrium.
In summary, we believe that although the environment of the embryo at the early stages of the study consisted mainly of tubal and uterine fluids, these fluid environments contained amino acids, proteins, and other substances. And it has been shown that cellular metabolism in both uterine and maternal plasma changes profoundly and more consistently during the preimplantation period. Therefore, we believe that the changes in the levels of the major compounds in the serum of the present study can reflect the changes in the levels of these compounds in the fallopian tubes and uterine cavity. The relevant discussion as well as references are located in lines 75-86.
Point 3: Authors should provide summary graphs or charts showing the dynamics of the serum levels of the main analyzed compounds.
Response 3: Dynamics of the serum levels of the main analyzed compounds in Table 5.(in line 347)
Point 4: The design of the figures could be more informative and user friendly. For example, instead of “Fig. 3-A1”, better “D1 vs D8 in ESI+ mode”, etc.
Response 4: All figures have been modified as required.
Point 5: Some files in supplementary materials duplicate the figures in the MS.
Response 5: Duplicate figures have been removed from the supplementary material and only the supplementary table has been retained.
Point 6: Line 21: Neovison vison should be italicized (it is a species name)
Response 6: Changes have been made in line 29. Neovison vison was modified to Neovison vison.
Point 7: Line 22: The incorrect statement “…the embryo is discharged from the uterus…” Embryos in a state of diapause are located in the uterine cavity.
Response 7: Changes have been made in lines 30-31. Mink embryonic diapause occurs during the blastocyst stage, a period of relatively static developmental state in embryonic development and metabolism.
Point 8: Line 34: The incorrect statement “…there is almost no mitosis or protein synthesis…” Indeed, the metabolic activity of embryos in the state of diapause is reduced; however, a basic level of protein synthesis is required to maintain the viability of embryos.
Response 8: Changes have been made in lines 40-42. The embryo's metabolism is greatly reduced during diapause, only basic protein synthesis and almost no mitosis. The embryo is thus in a free state within the uterus.
Point 9: Line 74: The characteristics of the experimental animals should be specified (age, body weight).
Response 9: Relevant information has been added in lines 95-96.
Point 10: Line 205: The incorrect statement “Most of the differentially-expressed amino acids…” The term “differential expression” applies only to genes.
Response 10: Changes have been made in line 217.Most of the differential amino acids annotated were down-regulated …

Round 2
Reviewer 1 Report
No special comments
it should be improved
Reviewer 2 Report
The authors improved the MS to a certain extent. However, further clarifications are required before publication of the paper.
1. It is still not clear how the experimental points correlate with the stages of mink embryo development. This primarily concerns to D22. The authors consider this point as a reactivation stage. However, they previously wrote: «Minks mate on March 15th and give birth on April 26th to produce offspring» (line 97). Thus, the total duration of pregnancy was about 42 days. Given that the post-implantation period is about 30 days in the mink, the D22 cannot correspond to reactivation, since reactivation and implantation should have been completed earlier. Authors should check and, if necessary, correct the data in lines 302-303 and/or 97.
2. Line 29: authors should use the term morula instead of “mulberry embryo”.